# Screening of Palladium/Charcoal Catalysts for Hydrogenation of Diene Carboxylates with Isolated-Rings (Hetero)aliphatic Scaffold

**DOI:** 10.3390/molecules28031201

**Published:** 2023-01-26

**Authors:** Vladyslav V. Subotin, Bohdan V. Vashchenko, Vitalii M. Asaula, Eduard V. Verner, Mykyta O. Ivanytsya, Oleksiy Shvets, Eugeniy N. Ostapchuk, Oleksandr O. Grygorenko, Sergey V. Ryabukhin, Dmitriy M. Volochnyuk, Sergey V. Kolotilov

**Affiliations:** 1Enamine Ltd., Chervonotkatska Street 78, 02094 Kyiv, Ukraine; 2L.V. Pisarzhevskii Institute of Physical Chemistry, National Academy of Sciences of Ukraine, Prosp. Nauky 31, 03028 Kyiv, Ukraine; 3Faculty of Chemistry, Taras Shevchenko National University of Kyiv, Volodymyrska Street 60, 01601 Kyiv, Ukraine; 4Institute of High Technologies, Taras Shevchenko National University of Kyiv, Volodymyrska Street 60, 01601 Kyiv, Ukraine; 5Institute of Organic Chemistry, National Academy of Sciences of Ukraine, Murmanska Street 5, 02660 Kyiv, Ukraine

**Keywords:** palladium, composite, catalysis, hydrogenation, hydrogen, diene, activity

## Abstract

A series of seven palladium-containing composites, i.e., four Pd/C and three Pd(OH)_2_/C (Pearlman’s catalysts), was prepared using modified common approaches to deposition of Pd or hydrated PdO on charcoal. All the composites were tested in the catalytic hydrogenation of diene carboxylates with the isolated-ring scaffold, e.g., 5,6-dihydropyridine-1(2*H*)-carboxylates with 2-(alkoxycarbonyl)cyclopent-1-en-1-yl and hex-1-en-1-yl substituents at the C(4)-position. The performance of the composites was also studied via the hydrogenation of quinoline as a model reaction. The composites were characterized by transmission and scanning electron microscopy (TEM and SEM), powder X-ray diffraction, and low-temperature N_2_ adsorption. It was found that the composites containing Pd nanoparticles (NPs) of 5–40 nm size were the most efficient catalysts for the hydrogenation of dienes, providing the reduced products with up to 90% yields at p(H_2_) = 100 atm, T = 30 °C for 24 h. The method of Pd NPs formation had more effect on the catalyst performance than the size of the NPs. The catalytic performance of Pearlman’s catalysts (Pd(OH)_2_/C) in the hydrogenation of dienes was comparable to or lower than the performance of the Pd/C systems, though the Pearlman’s catalysts were more efficient in the hydrogenation of quinoline.

## 1. Introduction

Catalytic hydrogenation of organic compounds is among the most important direct, cost-efficient and robust tools of classical and modern organic synthesis, which is widely applied both on micro- and multigram scales to produce pharmaceuticals, drug candidates, agrochemicals, building blocks, bulk solvents, etc. [1,2,3,4,5]. Platinum group metals (PGM), e.g., Pd- or Pt-nanoparticles deposited on various carriers, have been recognized as the most efficient catalysts due to their high activity and versatility for a range of transformations [6,7,8,9]. In contrast to other reducing agents, the advantage of molecular hydrogen is the green chemical aspect related to the production of a minimal quantity of waste [10]. However, despite the long history and wide use of PGM catalysts, the selection of the most effective system for certain reactions is not obvious in many cases, and quite often, the researchers use excessive loading of the catalysts in non-optimized reaction conditions with the aim of achieving a high yield of the hydrogenation product. Comparisons of the catalytic activity of the composites, which are prepared by different routes, and optimizations of the reaction conditions are highly demanded for efficient high-yielding productions. Several studies were devoted to the screening of the conditions and catalysts for the preparation of the most common organic compounds [10,11,12]. In turn, a lack of information about the optimized conditions for catalytic hydrogenation is observed within the novel and emerging chemotypes, i.e., functionalized conjugated dienes.

The latter reaction could provide sophisticated structural motifs and sp^3^-enriched compounds for modern drug design. For example, saturated scaffolds with isolated (hetero)cycles are promising three-dimensional motifs, which are widely represented in pharmaceuticals (Figure 1). The most prominent examples included rociverine [13,14], dicycloverine (dicyclomine) [15], benzetimide (levetimide and dexetimidum) [16,17,18], and fosinopril (monopril) [19,20,21]. From the synthetic chemists’ point of view, the preparation of a scaffold with isolated saturated rings might be challenging due to the required formal C(sp^3^)–C(sp^3^) coupling connectivity.

The widely used and versatile C(sp^2^)–C(sp^2^) coupling to synthesize dienes, followed by their exhaustive hydrogenation could be considered to be an alternative approach toward saturated ring-isolated (hetero)cycles. In turn, a lot of efforts were applied to selective mono-reduction of one double bond of dienes and polyenes [22,23,24,25]. Notably, no optimized reaction conditions and catalysts providing high reproducibility of methods have been reported to date for these molecular patterns. Moreover, hydrogenation of the conjugated dienes is not a trivial task because of their additional stabilization by the energy of conjugation [26]; therefore, this process often requires more harsh conditions as compared to hydrogenation of alkenes, nitro- or carbonyl compounds, etc.

In this study, we have aimed at the screening of a series of Pd-catalysts upon the common reaction conditions to find a well-tuned activity-loading pattern for the case of two representative examples of dienes, i.e., *N*-Boc protected 5,6-dihydropyridine-1(2*H*)carboxylates **1** and **2** bearing additional 2-(alkoxycarbonyl)-cyclopent-1-en-1-yl and hex-1-en-1-yl substituents at C(4)-position, respectively (Figure 2). Well-known protocols used for the preparation of Pd-containing hydrogenation catalysts were selected in order to evaluate the performance of the commonly used Pd-containing composites in this reaction. The performance of these catalysts was also evaluated by the hydrogenation of quinoline—a well-studied reaction, which can be proposed as a benchmark for the comparison of the catalytic performance in the hydrogenation reaction [11,12,27,28,29,30].

## 2. Results and Discussion

In this study, a series of palladium composites containing metallic Pd and so-called “palladium hydroxide” (Pearlman’s catalysts) on charcoal was prepared using a slight modification of the well-known procedures on a multigram scale (the composited are referred to as Pd/C and Pd(OH)_2_/C, respectively) [31,32,33]. Some reaction conditions in the protocols were slightly changed in order to study the influence of such deviations on the performance of the catalysts thus obtained. 

### 2.1. Synthesis and Characterization of Pd/C Composites

Synthesis of the four composites Pd/C-1–Pd/C-4 was carried out via reduction of the Pd^2+^ with formaldehyde in alkaline medium, which is a well-known approach to the formation of Pd/C catalysts [34]. 

In the case of Pd/C-1 and Pd/C-2, a solution of H_2_PdCl_4_ (0.0172 M for Pd/C-1 and 0.0155 M for Pd/C-2) was added to a suspension of the activated carbon and stirred overnight for complete adsorption of the Pd species on the carrier. The difference in the concentration of H_2_PdCl_4_ solutions could have an influence only on the sorption kinetics at the beginning of the process, because the Pd content in the samples was the same (within the determination error, Appendix A). Then, the activated carbon, which bore the adsorbed Pd species, was treated with formaldehyde; pH was adjusted identically for both composites via NaOH. Total palladium loading was 10% by weight (calculated for metal).

Nanocomposites Pd/C-3 and Pd/C-4 were obtained with inverted order of addition of the reagents, as compared to the case of Pd/C-1 and Pd/C-2. The procedure started from the addition of 30% aq NaOH into a suspension of charcoal in H_2_O, followed by sequential addition of PdCl_2_ in HCl and formaldehyde. Such sequence implies deposition of hydrated Pd^2+^ oxide (so-called “palladium hydroxide”) on the surface of the carrier, followed by its reduction to metallic Pd. In this case, the growth of Pd crystals had to be limited due to a lack of Pd^2+^ ions in the crystal growth area, which could not be supplied from the solution or could not migrate due to desorption from the charcoal. The resulting composites Pd/C = 3 and Pd/C-4 differ by Pd loading (10% and 20%, respectively). 

Powder X-ray diffraction (XRD) patterns of Pd/C composites are quite typical for nanostructured metal-containing systems [35]. There were distinct broad reflections on the powder XRD patterns (Figure 3), all of which could be assigned to metallic Pd with cubic unit cell [36] (typical hkl indexes are shown in brackets in Figure 3); wide halos centered at 2*θ* = 24° and 42° originate from the BAU charcoal. Thus, it can be concluded that the composite contained only metallic Pd as a crystalline phase.

The large width of the reflections of Pd can be an indication of the small size of the coherent scattering areas. The size of Pd crystallites, as estimated from the Sherrer equation [37], was ca. 30 nm for Pd/C-1, ca. 8 nm for both Pd/C-2 and Pd/C-3, and ca. 13 nm for Pd/C-4. 

For analysis of the size distribution of palladium nanoparticles, as well as their localization on the carrier, the composites obtained were analyzed by TEM (transmission electron microscopy) along with measurement of electron diffraction patterns and SEM (scanning electron microscopy). 

Distinct metallic nanoparticles (NPs) were revealed by TEM in the case of all four composites Pd/C-1, Pd/C-2, Pd/C-3, and Pd/C-4 (Figure 4). The shape of the NPs was irregular and can be considered to be close to spherical; the size of the NPs was in the range of 5–40 nm for Pd/C-1, 5–20 nm for Pd/C-2, and 5–15 nm for both Pd/C-3 and Pd/C-4. The sizes estimated from TEM images are consistent with the results of XRD analysis: in particular, large Pd particles were detected in the case of Pd/C-1 by both methods, and XRD did not indicate the presence of large (several dozen nm) particles in the cases of other composites, and such particles indeed were not found by TEM.

Single reflections were observed on the electron diffraction pattern of Pd/C-2 (Figure 4d). It can be concluded that this sample apparently contained some quantity of large crystals of Pd (this finding does not contradict the simultaneous presence of small Pd NPs). However, the number of such particles was not high, because they did not make a significant contribution to the XRD reflection width, and the result of the electron diffraction experiment could be considered to be an indication of the non-homogeneity of the composite. Bright concentric electron diffraction signals were observed in the case of the Pd/C-4 composite (Figure 4h), thus providing evidence for the significant contribution of relatively large Pd crystals. Notably, Pd content in Pd/C-4 was 20%, which is two times higher than in Pd/C-1, Pd/C-2 and Pd/C-3l; therefore, more intense electron diffraction can be explained, in part, by higher Pd content. In turn, large crystals were not detected in the case of Pd/C-1 and Pd/C-3 by electron diffraction.

SEM provides information about the topography of the composite (secondary electrons mode) and phase composition (back-scattered electrons mode, which indicates the bright-looking Pd-containing phase on the dark-colored remaining phases, which contain only light elements). Analysis of SEM images of Pd/C composites revealed large agglomerates in the case of Pd/C-1 and Pd/C-2 composites, with linear sizes of up to 300 nm and 800 nm, respectively (Figure 5). The aggregates of Pd in the case of Pd/C-3 were much smaller (mostly of sizes of less than 200 nm); uniform distribution of these agglomerates on the surface of the composite was also observed. Significantly more regular distribution of Pd on the surface of the carrier was found for the case of Pd/C-4, which is especially remarkable in view of larger Pd content, i.e., 20% instead of 10%, vide supra (increase of Pd content could lead to formation of large metallic particles, which, however, were not observed).

Generally, all Pd/C composites studied had a rather wide distribution of Pd NPs by size, which is a common result for the case of preparation of the composites by “simple” Pd^2+^ reduction without surfactants or special reagents for size control of NPs. However, it can be noted that even within the method used, variation of the deposition conditions of Pd NPs had a significant effect on the overall structure of the nanocomposite. Adsorption of H_2_PdCl_4_ followed by reduction with alkaline formaldehyde led to the formation of Pd NPs, which formed agglomerates. In this case, a lower concentration of H_2_PdCl_4_ at the first stage of adsorption led to the growth of larger NPs, probably due to the formation of larger crystal growth seeds [38,39] in conditions of adsorption closer to equilibrium. In turn, such larger NPs promote higher agglomeration of Pd on the surface. On the other hand, the reduction of the hydrated Pd^2+^ oxide, which was deposited on the charcoal (the result of the addition of a Pd^2+^ source to the charcoal, impregnated with NaOH) led to more even distribution of the Pd NP, probably due to the limited diffusion capacity of Pd^2+^ in the process of the NPs growth. 

### 2.2. Synthesis and Characterization of Pearlman’s Catalysts

Considering all advantages of using PdO/OH/H_2_O/C (the simplified structural representation “Pd(OH)_2_/C”) as a pre-catalyst for in situ generations of metal Pd(0)-NPs in the reaction mixture [40,41], we have aimed at the synthesis of three samples by modification of the known literature procedure to compare the relative catalytic activity [33]. The method included treatment of the activated charcoal with Na_2_CO_3_ in different concentrations followed by the addition of H_2_PdCl_4_. Three samples Pd(OH)_2_/C-1–3 were obtained with a shifted *χ*(Na/Pd) ratio from ca. 9 to ca. 21 and pH value from 10 to 12 in order to evaluate the possible influence of the preparation method of the nanocomposites on their catalytic properties.

XRD analysis of the Pd(OH)_2_/C composites revealed that these samples did not contain phases with high crystallinity. Remarkably wide reflections were found on the XRD patterns, which could be assigned to BAU carrier (2*θ* = 24° and 42°, the same, as in the case of Pd/C composites) and to PdO with tetragonal unit cell (Figure 6) [42]. The size of PdO crystallites, as estimated from Sherrer’s equation, was less than 3 nm. No reflections which could be associated with metallic Pd were found in the case of Pd(OH)_2_/C composites. 

Because all samples of the Pd(OH)_2_/C series which were studied here had similar XRD diffraction and similar catalytic performance in the hydrogenation of both quinoline and dienes (vide infra), TEM and SEM studies were carried out for one typical representative of this row, i.e., sample Pd(OH)_2_/C-2. No distinct NPs were found in the Pd(OH)_2_/C-2 composite by TEM (Figure 7a), and no signs of single crystals were revealed by electron diffraction (Figure 7b). 

These results are consistent with the XRD pattern of this composite, which could indicate that the crystallinity of the sample was very poor. The results of XRD analysis and TEM examination of the Pd(OH)_2_/C-2 composite are completely consistent with the previously reported data [43].

The SEM images of the Pd(OH)_2_/C-2 composite are presented in Figure 8. It could be concluded from SEM analysis that hydrated PdO covered the particles of the charcoal with an almost less uniform layer (Figure 8 and Appendix A). Large particles, which could be associated with pure PdO, were not found. 

### 2.3. Characteristics of the Porous Structure of the Composites

Because the size of the Pd NPs, which was evaluated by TEM and SEM, was larger than the size of pores of BAU charcoal (BAU charcoal is a microporous sorbent), it could be expected that Pd NPs were localized not in the pores of the carrier. In order to verify this supposition and for characterization of the change of porosity of the systems upon deposition of Pd, N_2_ sorption measurements were carried out for selected samples (Figure 9). 

The values of both specific surface area and pore volume of the composites quite monotonously decreased, as compared to starting BAU carrier, upon deposition of Pd or PdO (Table 1). Such decrease could be explained first of all by the “addition” of 10–20% of dense Pd, which expectedly led to a 10–20% decrease in the sorption characteristics of the carrier via the dilution of porous material with non-porous additive, providing a proportional decrease of the surface area and pore volume (per 1 g of the system). However, some important notes can be made. First, the values of the specific surface in the cases of Pd/C-1 and Pd(OH)_2_/C-2 composites decreased much more significantly, when compared to the starting carrier, than the expected decrease at the level of 10% or 20%, respectively (which correspond to weight % of Pd loading in Pd/C-1 and Pd(OH)_2_/C-2). In both cases, this change could be an indication of partial blocking of the pores by Pd-containing species. Second, the diameter of the pores did not decrease significantly in all cases, and this observation can indicate that there are no sub-nanoparticles located directly in pores, except those blocking the pore completely (the latter cause decrease of the pore volume with retained measured pore diameter). It can be concluded that the accessible Pd NPs are located on the BAU particles only on the external surface or over the pore “window”. Thus, it can be concluded that the hydrogenation process takes place on the Pd NPs, which is localized on the surface of BAU particles (in contrast to the reaction inside the pores of the BAU carrier). 

### 2.4. Catalytic Performance of the Composites in the Hydrogenation of Quinoline and Dienes

The catalytic properties of the composites were tested in the reaction of quinoline hydrogenation because this reaction was widely used for the evaluation of the hydrogenation catalysts [11,12,27,28,29,30]. The reaction conditions (T = 50 °C, p(H_2_) = 30 bar, t = 4 h, Table 2) were specially selected in such a manner that complete conversion of quinoline to 1,2,3,4-tetrahydroquinoline was not achieved (Table 2). Such an experiment allowed ranging of the performance of these samples. Clearly, increase of the reaction temperature or pressure or time to certain value would lead to the complete conversion of the quinoline in all these cases, but the comparison of the catalytic performance of the samples would not be possible. 

Samples Pd(OH)_2_/C-1, Pd(OH)_2_/C-2, and Pd(OH)_2_/C-3 had comparably similar performance in the hydrogenation of quinoline: the yields of 1,2,3,4-tetrahydroquinoline were in the range of 55–67%. Pd/C samples appeared to be less active, among them Pd/C-1 showed the best result (42% yield of 1,2,3,4-tetrahydroquinoline), despite the largest size of Pd NPs in the row of Pd/C composites which are considered in this study. At the same time, performance of Pd/C-3 and Pd/C-4 composites in the hydrogenation of quinoline was the lowest (*ca.* 10% yield of 1,2,3,4-tetrahydroquinoline). It should be noted that the size of Pd NPs was not the sole factor that determined their catalytic activity. Furthermore, deposition of hydrated PdO followed by in situ reductions with H_2_ (changes of Pd(OH)_2_ system caused by hydrogenation were shown by TEM, Appendix A) provided the most efficient composites, whereas deposition of hydrated PdO followed by reduction with formaldehyde led to the formation of the least active systems. Though the majority of Pd/Support composites contain Pd in the oxidation states 0 and +2 simultaneously [44,45,46,47,48], the action of hydrogen at the beginning of the hydrogenation process results in a reduction of Pd^2+^ [44,49] (similar reduction of Pd^2+^ to Pd^0^ was observed under the action of CO [50]). Thus, different activity of the composites can be explained by in situ formation of the NPs, which have a different shape (different crystallographic planes exposed to the surface) or content of defects. 

The catalysts Pd/C-1–4, and Pd(OH)_2_/C-1–3 were then tested in the diene hydrogenation, and their performance was compared with commercially available catalysts 5% Pd/C and 20% Pd(OH)_2_/C, widely used in our synthetic projects (Figure 10, Table 2). The experiments were carried out using 10 mol. % loading of Pd in the case of all composites, p(H_2_) = 100 atm, T = 30 °C for 24 h. As compared to quinoline, significantly higher Pd loading was required in the case of hydrogenation of dienes to achieve high yields; this difference might be addressed to the expected lower activity of dienes towards H_2_ addition. A low temperature (30 °C) was selected to minimize the formation of the by-products.

Initially, commercially available 5% Pd/C was tested in the processes of hydrogenation of dienes **1** and **2** in different solvents, i.e., non-polar aprotic hexanes (Table 3, Entries 1 and 12), polar aprotic THF (Entries 2 and 13), and polar protic MeOH (Entries 3 and 14). Although using the first two solvents provided the target products **3** and **4** only in up to 4% yield, a dramatic increase of up to 21% and 56%, respectively, was observed for hydrogenation in the MeOH medium. These results are in accordance with the solubility of molecular H_2_, which increases from THF via hexane to the highest values in MeOH [51,52]. Therefore, all further experiments with synthesized catalysts were performed in MeOH. Notably, the hydrogenation of diene **2** with a six-membered ring proceeded with a higher yield compared to the five-membered analog **1**, which might be attributed to the effect of ring size.

Among the Pd/C systems prepared in this study, the best results were observed in the case of Pd/C-1 (Entries 4 and 15) and Pd/C-2 (Entries 5 and 16). In turn, the composites Pd/C-3 and Pd/C-4 were not active in the hydrogenation of diene **1** (Entries 6 and 7), whereas the hydrogenation of diene **2** resulted in a 24–26% yield of **4** (Entries 17 and 18). These findings agree with the difference in the yields of 1,2,3,4-tetrahydroquinoline obtained in the presence of the same composites Pd/C-1–Pd/C-4. Because the performance of the composites in different reactions was similar, it can be supposed that different activities of the catalysts in this Pd/C series was associated with the state of Pd in composite, in contrast to possible specific reaction selectivity. 

The use of Pd(OH)_2_ composites for hydrogenation of diene **1** (less reactive than **2**) led to the formation of **3** with yields, comparable to the case of Pd/C-1 and Pd/C-2 composites (the yields were in the range 21–35%; 33–35% in the majority of cases). However, the performance of Pd/C-1 and Pd/C-2 in the case of more active diene **2** was significantly higher (Entries 15 and 16), than the performance of Pd(OH)_2_/C systems (Entries 19–21). These results contradict the performance of these composites in the hydrogenation of quinoline, for which Pd(OH)_2_/C systems showed better results compared to the experiments with their Pd/C counterparts. The disagreement between the productivity of the composites in the hydrogenation of quinoline and dienes can be explained by the difference of certain reaction stages (such as different energies of adsorption of quinoline and dienes on the catalyst), as well as the formation of different catalytically active sites in the presence of these substrates [53]. Nevertheless, all catalysts Pd/C-1–Pd/C-4 and Pd(OH)_2_/C-1–Pd(OH)_2_/C-3 were more active compared to commercially available analogs 5% Pd/C (Entries 3 and 14) and 20% Pd(OH)_2_/C (Entries 11 and 22). 

It should be mentioned that no products of partial hydrogenation of diene **1** were observed by using either Pd/C-1 or Pd/C-2. It can be supposed that hydrogenation of the possible intermediates (the compounds bearing one C=C bond) on these catalysts occurs easier (quicker) than hydrogenation of the starting dienes. Formation of the products of partial hydrogenation in other cases (entries 6, 7, 17–21) can be explained by comparable rates of hydrogenation of the starting compounds and of the intermediates. 

## 3. Materials and Methods

The solvents were purified according to standard procedures [54]. Hydrogen (99.99%) was purchased from Galogas Ltd. (Kyiv, Ukraine) and was used without further purification. Commercially available 5% Pd/C and 20% Pd(OH)_2_/C were purchased from Daming Ruiheng Chemical Co. Ltd. (Hebei, China). All other starting materials and reagents were available from Enamine Ltd. (Kyiv, Ukraine) and UkrOrgSintez Ltd. (Kyiv, Ukraine) Commercially available activated crushed porous charcoal “BAU-A” (Ukrainian state standard 6217-74) was used; the charcoal was prepared from environmentally friendly raw materials (birch wood) upon treatment of water vapor at a temperature of 800–950 °C, followed by crushing. For this study, the commercially available activated charcoal was initially processed by heating on a steam bath with 10% HNO_3_ for 2–3 h, then washed to a free-of-acid state with H_2_O, and dried at 100–110 °C before use. TEM and electron diffraction measurements were performed using a PEM-125K instrument (SELMI, Sumy, Ukraine) operating at 100 kV acceleration voltage. Samples were suspended in methanol upon ultrasonic irradiation for 1 min, and a drop of the suspension was applied onto Cu grid (300 mesh), which was covered by a film of amorphous carbon, immediately after the end of the ultrasonic treatment. SEM measurements were carried out using a Carl Zeiss NVision 40 scanning electron microscope (micrographs were obtained at 7 kV acceleration voltage). The samples were not specially prepared (in particular, they were not coated with conducting material) for SEM measurements. Pd content in the composites was determined by atomic adsorption using an Thermo Scientific iCE3500 spectrometer (Thermo Fisher Scientific Inc., Waltham, MA, USA) with acetylene-air flame atomization. Prior to analysis, samples were dried in vacuum 10^−4^ Torr at 110 °C for 24 h in order to remove water adsorbed from the air. Powder X-ray diffraction measurements were performed using a Bruker D8 Advance diffractometer with CuKα radiation (λ = 1.54056 Å). Measurements of N_2_ sorption were studied on a Sorptomatic-1990 instrument by volumetric method at 78 K. Prior to measurements samples were heated at 200 °C in 10^−4^ Torr vacuum for 2 h.

Column chromatography was performed using Kieselgel Merck 60 (230–400 mesh) as the stationary phase. ^1^H and ^13^C{^1^H} spectra were recorded on an Agilent ProPulse 600 spectrometer (at 151 MHz for ^13^C NMR), a Bruker 170 Avance 500 spectrometer (at 500 MHz for ^1^H NMR, 126 MHz for ^13^C{^1^H} NMR), and a Varian Unity Plus 400 spectrometer (at 400 MHz for ^1^H NMR, 101 MHz for ^13^C{^1^H} NMR). NMR chemical shifts are reported in ppm (δ scale) downfield from TMS as an internal standard and are referenced using residual NMR solvent peaks at 7.26 and 77.16 ppm for ^1^H and ^13^C{^1^H} in CDCl_3_, 2.50 and 39.52 ppm for ^1^H and ^13^C{^1^H} in DMSO-d_6_. Coupling constants (*J*) are given in Hz. Spectra are reported as follows: chemical shift (δ, ppm), multiplicity (s—singlet, t—triplet, q—quartet, quint.—quintet, dd—doublet of doublets, dt—doublet of triplets, td—triplet of doublets, quint.d,—quintet of doublets, m—multiplet), integration, and coupling constants (Hz). Elemental analyses were performed at the Laboratory of Organic Analysis, Department of Chemistry, Taras Shevchenko National University of Kyiv. Mass spectra were recorded on an Agilent 1100 LCMSD SL instrument (chemical ionization (CI)) and an Agilent 5890 Series II 5972 GCMS instrument (electron impact ionization (EI)). High-resolution mass spectra (HRMS) were recorded on an Agilent Infinity 1260 UHPLC system coupled to 6224 Accurate Mass TOF LC/MS system. Hydrogenation experiments were carried out in a steel autoclave equipped with a manometer, magnetic stirrer, and temperature controller.

Nanocomposites Pd/C-1 and Pd/C-2 were prepared by a slight modification of the known literature procedure [31]. The reaction scheme is based on the sequential addition as follows: C (H_2_O) + PdCl_2_ × HCl + CH_2_O + NaOH → Pd/C. The methods differ by the concentrations of the PdCl_2_ solution which was used for the reaction. 

Nanocomposites Pd/C-3 and Pd/C-4 were prepared by modification of the procedure known from the literature [32]. The reaction scheme is based on the sequential addition as follows: C (H_2_O) + NaOH + PdCl_2_ × HCl + CH_2_O → Pd/C.

Nanocomposites Pd(OH)_2_/C-1–3 were prepared according to modification of the procedure known from the literature [33]. The reaction scheme is based on the sequential addition as follows: C (H_2_O) + Na_2_CO_3_ + PdCl_2_×HCl → Pd(OH)_2_/C. The composites differ by the quantity of Na_2_CO_3_ added per 1 g of PdCl_2_.

Details of the preparation of the composites and general synthetic procedures for preparation of **1**–**4** are provided in the Appendix A.

Synthesis of *tert-Butyl 4-(2-(methoxycarbonyl)cyclopent-1-en-1-yl)-5,6-dihydro-pyridine-1(2H)-carboxylate (**1**).* The analytical sample was obtained by purification via column chromatography on silica gel using gradient hexanes—t-BuOMe as an eluent. Yellowish oil. The compound existed as a ca. 1:1 mixture of rotamers. ^1^H NMR (400 MHz, CDCl_3_) δ 5.64 (s, 1H), 4.02–3.91 (m, 2H), 3.68 (s, 3H), 3.48 (t, J = 5.6 Hz, 2H), 2.67 (t, J = 7.6 Hz, 2H), 2.59 (td, J = 7.6, 2.3 Hz, 2H), 2.29–2.17 (m, 2H), 1.85 (p, J = 7.6 Hz, 2H), 1.44 (s, 9H). ^13^C NMR (126 MHz, CDCl_3_) δ 166.7, 154.3, 152.0, 132.8, 127.6, 122.9 and 122.3, 79.1, 50.9, 43.2 and 42.6, 40.5 and 39.2, 37.7, 34.7, 28.0, 27.2, 21.2. LC/MS (ES-API): *m/z* = 308 [M + H]^+^. Anal. calcd. for C_17_H_25_NO_4_: C 66.43; H 8.20; N 4.56. Found: C 66.11; H 7.85; N 4.57.

Synthesis of *tert-Butyl 4-(2-(ethoxycarbonyl)cyclohex-1-en-1-yl)-5,6-dihydropyri-dine-1(2H)-carboxylate (**2**)*. The analytical sample was obtained by purification via column chromatography on silica gel using gradient hexanes–*t*-BuOMe as an eluent. Yellowish oil. The compound existed as a ca. 3:2 mixture of rotamers. ^1^H NMR (400 MHz, CDCl3) δ 5.30 (s, 1H), 4.11 (q, J = 7.1 Hz, 2H), 3.93–3.81 (m, 2H), 3.53 (t, J = 5.5 Hz, 2H), 2.29 (dt, J = 6.6, 3.7 Hz, 2H), 2.24–2.16 (m, 2H), 2.13 (dd, J = 6.6, 3.7 Hz, 2H), 1.68–1.57 (m, 4H), 1.47 (s, 9H), 1.22 (t, J = 7.1 Hz, 3H). ^13^C NMR (126 MHz, CDCl_3_) δ 169.2, 154.4 and 154.3, 146.2 and 146.0, 138.9 and 138.7, 125.8, 118.2 and 117.9, 79.0, 59.6, 42.9 and 42.4, 40.4 and 39.2, 29.6, 28.0, 27.3 and 27.2, 25.7, 21.6, 21.5, 13.8. LC/MS (ES-API): *m/z* = 336 [M + H]^+^. Anal. calcd. for C_19_H_29_NO_4_: C 68.03; H 8.71; N 4.18. Found: C 68.16; H 9.06; N 4.51.

Synthesis of *tert-Butyl 4-(2-(methoxycarbonyl)cyclopentyl)piperidine-1-carboxy¬late (**3**)*. The analytical sample was obtained by purification via column chromatography on silica gel using gradient hexanes—t-BuOMe as an eluent. The compound was obtained as a ca. 8:7 mixture of diastereomers. Yellowish oil. ^1^H NMR (400 MHz, CDCl3) δ 4.05 (t, J = 12.9 Hz, 2H), 3.64 (s, 1.4H) and 3.62 (s, 1.6H), 2.90 (t, J = 7.1 Hz, 0.53H) and 2.67–2.54 (m, 2H) and 2.51–2.36 (m, 0.47H), 2.03 (quint., J = 8.3 Hz, 0.47H) and 1.94–1.69 (m, 4.53H), 1.68–1.45 (m, 5H), 1.42 (s, 9H), 1.39–1.17 (m, 1.87H) and 1.16–1.01 (m, 2.13H). ^13^C NMR (126 MHz, CDCl_3_) δ 177.1 and 175.8, 154.4 and 154.3, 78.8 and 78.7, 51.2 and 50.3, 50.6 and 47.0, 48.6 and 45.2, 43.6 and 43.5 and 43.5 and 43.4, 40.4 and 37.6, 31.3 and 31.1 and 30.9, 30.3 and 29.8, 29.5 and 29.1, 28.0, 24.9 and 22.8. LC/MS (ES-API): *m/z* = 312 [M + H]^+^. Anal. calcd. for C_17_H_29_NO_4_: C 65.57; H 9.39; N 4.50. Found: C 65.40; H 9.53; N 4.10.

Synthesis of *tert-Butyl 4-(2-(ethoxycarbonyl)cyclohexyl)piperidine-1-carboxylate (**4**)*. The analytical sample was obtained by purification via column chromatography on silica gel using gradient hexanes—*t*-BuOMe as an eluent. The crude product was obtained as a ca. 9:2 mixture of cis- and trans-diastereomers. The analytical sample was obtained as a single *cis*-diastereomer. Yellowish oil. ^1^H NMR (400 MHz, CDCl_3_) δ 4.19–3.97 (m, 4H), 2.83–2.72 (m, 1H), 2.65–2.52 (m, 2H), 2.02–1.90 (m, 1H), 1.83–1.69 (m, 3H), 1.67–1.55 (m, 3H), 1.50–1.45 (m, 3H), 1.42 (s, 9H), 1.22 (t, J = 7.1 Hz, 3H), 1.16 (dt, J = 9.2, 4.2 Hz, 2H), 0.97 (quint.d, J = 12.4, 4.2 Hz, 2H). ^13^C NMR (126 MHz, CDCl_3_) δ 174.1, 154.3, 78.7, 59.2, 43.9, 43.5, 40.6, 37.8, 29.9, 29.6, 28.8, 28.0, 25.5, 24.6, 21.5, 13.8. LC/MS (ES-API): *m/z* = 340 [M + H]^+^. Anal. calcd. for C_19_H_33_NO_4_: C 67.22; H 9.80; N 4.13. Found: C 66.88; H 10.09; N 4.28.

Hydrogenation experiments. The organic compound (quinoline, diene **1** or **2**), palladium catalyst, and solvents were mixed directly in an autoclave, which was purged by argon. The autoclave was sealed, purged with hydrogen, and pressurized with hydrogen quickly after the mixing of the reagents. Time, temperature, and duration of the processes are indicated in the text of the paper. After completion of the reaction, the catalyst was separated by centrifugation, the reaction mixture was evaporated, and the organic residue was analyzed by NMR and LC/MS, as reported previously [28,29,30]. 

## 4. Conclusions

In this study, the performance of a series of Pd-catalysts which were prepared by well-known procedures was evaluated in the hydrogenation of the two model diene carboxylates and, for comparison, the hydrogenation of quinoline. It was found that the composites formed by adsorption of Pd^2+^ on the charcoal followed by reduction to metallic Pd had significantly higher performance in the hydrogenation of dienes and quinoline compared to the composites, which were prepared by precipitation of hydrated PdO on the charcoal followed by its reduction. The effect of the method of preparation of the composite on its performance was more significant than the effect of the size of Pd NPs or their distribution (agglomeration) on the surface of the carrier had on such performance. 

Catalytic performance of Pearlman’s catalysts (Pd(OH)_2_/C) in hydrogenation of dienes was comparable or lower than the performance of the Pd/C systems, whereas Pearlman’s catalysts were more efficient in the hydrogenation of quinoline. Such difference can be explained by the formation of different catalytically active metallic NPs in the course of hydrogenation of different organic substrates. 

It was established that the yield of the products in the hydrogenation of dienes increased in the following order of solvents: THF < hexanes << MeOH, which correlates with an increase in the solubility of H_2_. 

Finally, efficient catalysts for the hydrogenation of dienes were found among well-known systems. We believe that the results presented in this study will be suitable for other related reactions and substrates and can be useful for researchers working in the field of synthetic organic chemistry and catalysis.

## Figures and Tables

**Figure 1 molecules-28-01201-f001:**
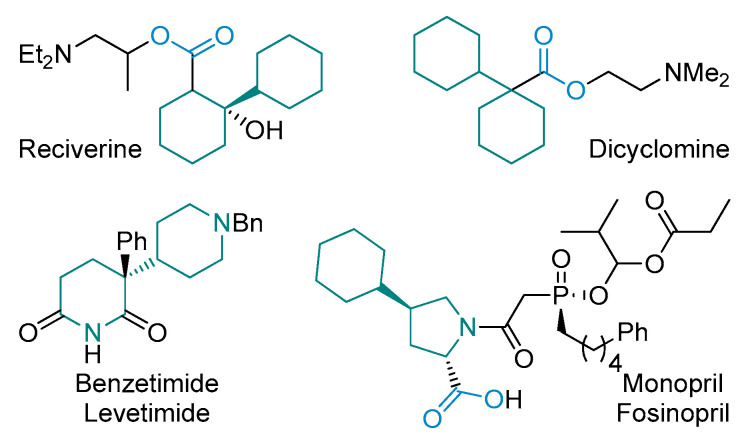
Pharmaceutically relevant examples of pharmaceuticals based on the scaffold with isolated (hetero)aliphatic rings.

**Figure 2 molecules-28-01201-f002:**
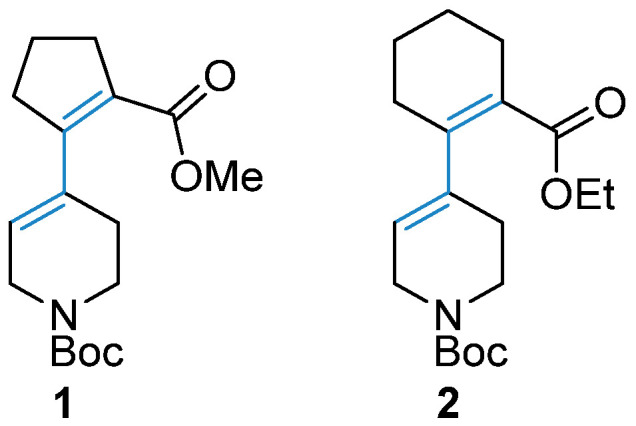
Formulae of conjugated dienes considered in this study.

**Figure 3 molecules-28-01201-f003:**
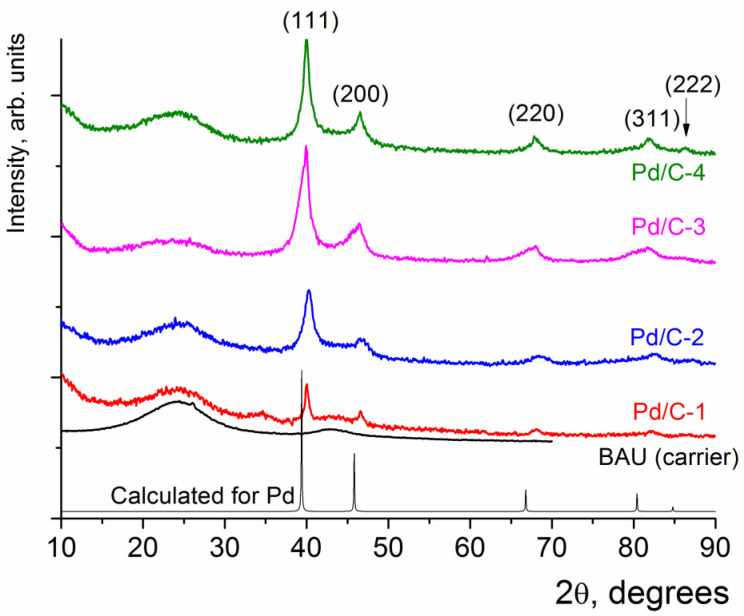
XRD patterns for Pd/C-1, Pd/C-2, Pd/C-3, and Pd/C-4 composites.

**Figure 4 molecules-28-01201-f004:**
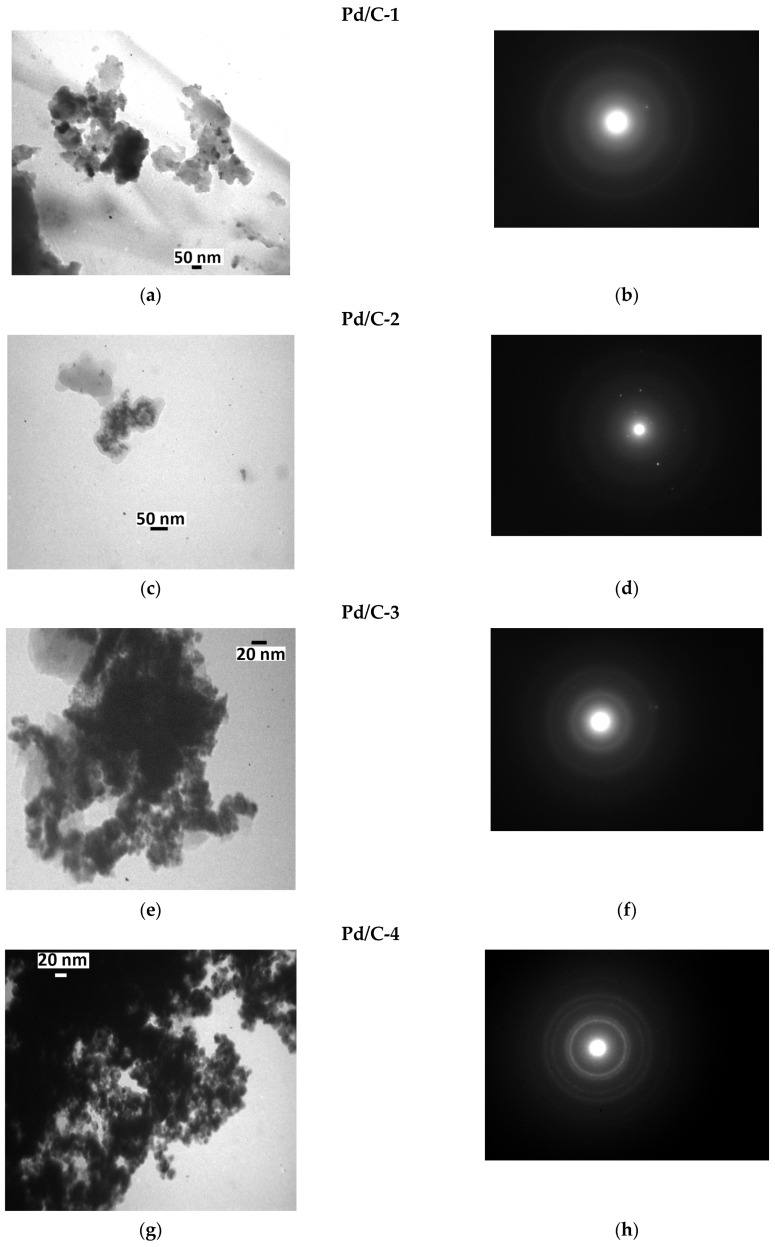
TEM images (left column) and electron diffraction patterns (right column) of Pd/C-1 (**a**,**b**), Pd/C-2 (**c**,**d**), Pd/C-3 (**e**,**f**), and Pd/C-4 (**g**,**h**) composites.

**Figure 5 molecules-28-01201-f005:**
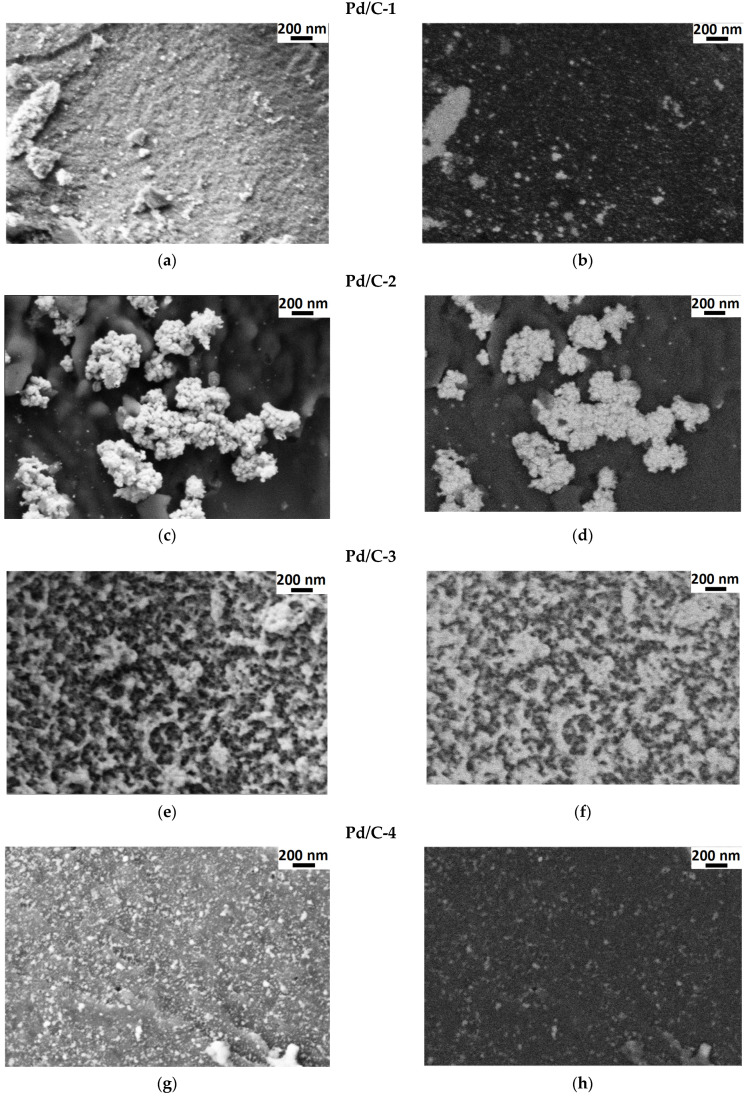
SEM images in secondary electron mode (left column) and backscattered electron mode (right column) of Pd/C-1 (**a**,**b**), Pd/C-2 (**c**,**d**), Pd/C-3 (**e**,**f**), and Pd/C-4 (**g**,**h**) composites.

**Figure 6 molecules-28-01201-f006:**
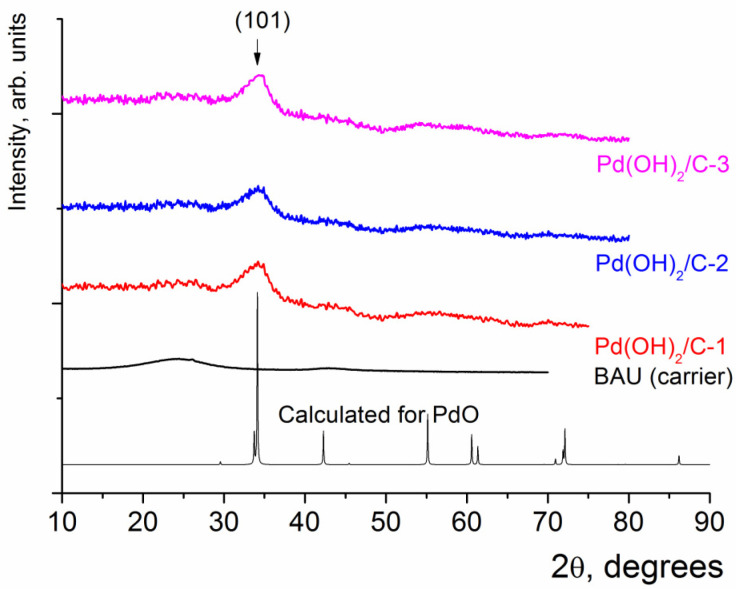
XRD patterns of Pd(OH)_2_/C composites.

**Figure 7 molecules-28-01201-f007:**
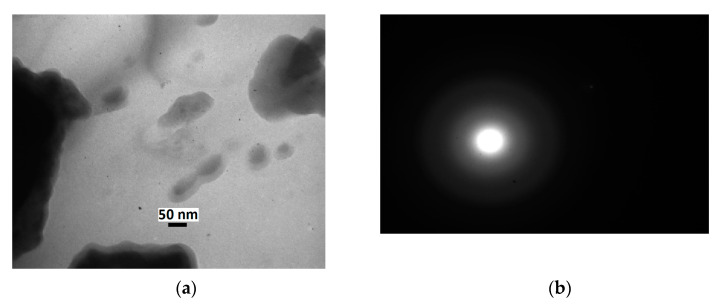
TEM image (**a**) and electron diffraction pattern (**b**) of Pd(OH)_2_/C-2.

**Figure 8 molecules-28-01201-f008:**
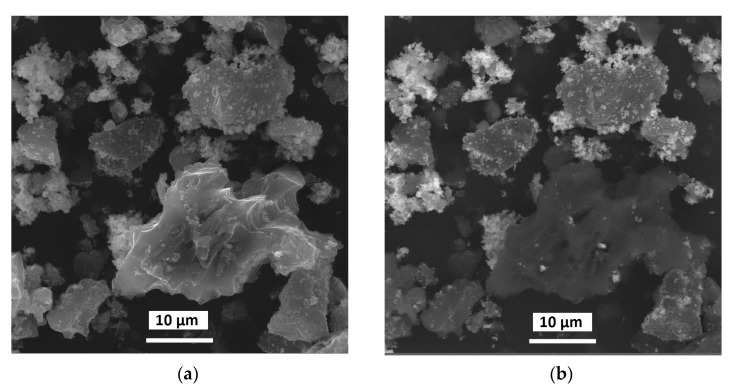
SEM images in secondary electrons mode (**a**) and backscattered electron mode (**b**) for Pd(OH)_2_/C-2.

**Figure 9 molecules-28-01201-f009:**
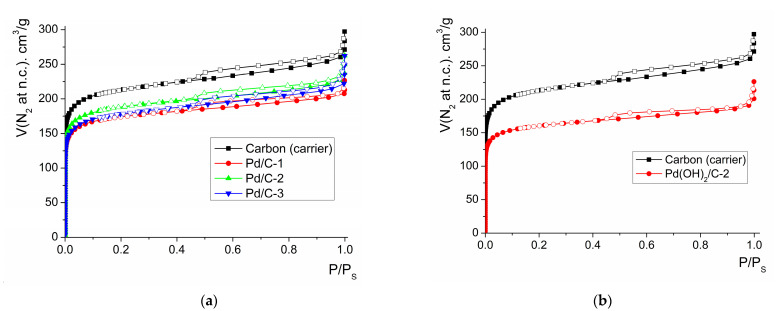
Isotherms of N_2_ adsorption (filled points) and desorption (empty points) for Pd/C composites (**a**) and Pd(OH)_2_/C-2 composite (**b**) at 78 K.

**Figure 10 molecules-28-01201-f010:**
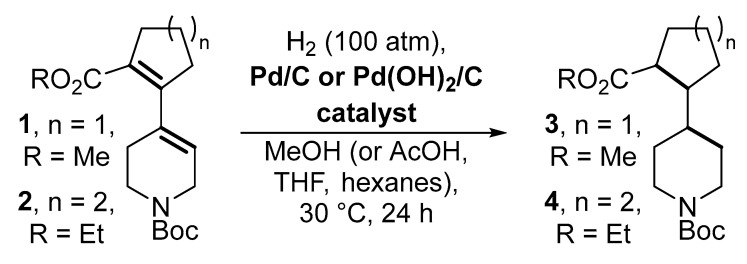
Scheme of the catalytic hydrogenation of *N*-Boc protected 5,6-dihydro-pyridine-1(2*H*)-carboxylates **1** and **2** bearing additional 2-(alkoxycarbonyl)-cyclopent-1-en-1-yl and hex-1-en-1-yl substituents at C(4)-position, respectively.

**Table 1 molecules-28-01201-t001:** Surface area and pore volume of the composites and starting carrier calculated from N_2_ adsorption at 77 K.

Sample	S_BET_ ^a^, m^2^/g	V_T_, cm^3^/g	V_micro_, cm^3^/g	D_micro_, nm
Charcoal (carrier)	825	0.395	0.302	0.569
Pd/C-1	580	0.313	0.243	0.566
Pd/C-2	635	0.344	0.264	0.560
Pd/C-3	690	0.331	0.249	0.570
Pd(OH)_2_/C-2	623	0.290	0.226	0.554

^a^ S_BET_ is a specific surface by Brunauer–Emmett–Teller; V_T_ is total pore volume by Gurvich at P/P_S_ = 0.95; V_micro_ is the specific volume of micropores by Dubinin and Raduskevich; D_micro_ is the median pore diameter by Horvath and Kawazoe.

**Table 2 molecules-28-01201-t002:** Yields of 1,2,3,4-tetrahydroquinoline in presence of the composites (T = 50 °C, p = 30 bar, t = 4 h, methanol was used as solvent). Pd loading in the reaction mixture was 1 mol. % in all cases.

Catalyst	Yield ^a^, %
Pd/C-1	42
Pd/C-2	28
Pd/C-3	10
Pd/C-4	9
Pd(OH)_2_/C-1	61
Pd(OH)_2_/C-2	67
Pd(OH)_2_/C-3	55

^a^ Besides 1,2,3,4-tetrahydroquinoline, the reaction mixture contained mainly unreacted quinoline and less than 1% of other products (first of all *N*-methyl-1,2,3,4-tetrahydroquinoline).

**Table 3 molecules-28-01201-t003:** Catalytic hydrogenation of dienes **1** and **2**: screening of Palladium catalysts.

No.	Diene	Palladium Catalysts	Pd Loading, mol. %	Solvent	Yield of 3 or 4 (%) ^a^
1	**1**	5% Pd/C (commercial)	10	hexanes	3.5
2	**1**	5% Pd/C (commercial)	10	THF	3.5
3	**1**	5% Pd/C (commercial)	10	MeOH	21
4	**1**	10% Pd/C-1	10	MeOH	35
5	**1**	10% Pd/C-2	10	MeOH	35
6	**1**	10% Pd/C-3	10	MeOH	0, N/D ^b^ (12% partially reduced products)
7	**1**	20% Pd/C-4	10	MeOH	0, N/D ^b^ (18% partially reduced products)
8	**1**	20% Pd(OH)_2_/C-1	10	MeOH	33
9	**1**	20% Pd(OH)_2_/C-2	10	MeOH	35
10	**1**	20% Pd(OH)_2_/C-3	10	MeOH	27
11	**1**	20% Pd(OH)_2_/C (commercial)	10	MeOH	20
12	**2**	5% Pd/C (commercial)	10	hexanes	0, N/D ^b^
13	**2**	5% Pd/C (commercial)	10	THF	1.3
14	**2**	5% Pd/C (commercial)	10	MeOH	56
15	**2**	10% Pd/C-1	10	MeOH	92
16	**2**	10% Pd/C-2	10	MeOH	89
17	**2**	10% Pd/C-3	10	MeOH	24 (59% partially reduced products)
18	**2**	10% Pd/C-4	10	MeOH	26 (58% partially reduced products)
19	**2**	20% Pd(OH)_2_/C-1	10	MeOH	66 (19% partially reduced products)
20	**2**	20% Pd(OH)_2_/C-2	10	MeOH	64 (12% partially reduced products)
21	**2**	20% Pd(OH)_2_/C-3	10	MeOH	55 (15% partially reduced products)
22	**2**	20% Pd(OH)_2_/C (commercial)	10	MeOH	51

^a^ Yield determined by LC/MS. In the case when only the yield of **3** or **4** is indicated, no by-products were found. ^b^ N/D—product not detected.

## Data Availability

The data presented in this study are available on request from the corresponding author.

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
