# Peer review of "Screening of Palladium/Charcoal Catalysts for Hydrogenation of Diene Carboxylates with Isolated-Rings (Hetero)aliphatic Scaffold"

_molecules, 2023, doi:10.3390/molecules28031201_

Round 1

Reviewer 1 Report

In the present manuscript entitled “Screening of Palladium/Charcoal Catalysts for Hydrogenation of Diene Carboxylates with Isolated-Rings (Hetero)aliphatic Scaffold” authors have reported a series of Pd-carbon based catalysts, prepared by known procedures, for selective hydrogenation of diene carboxylates and, quinoline. The authors characterized the synthesized catalysts using various tools; and studied the hydrogenation reaction in-depth by varying reaction parameters. However, the manuscript still requires major modifications before acceptance for publication:

1. The sub-sections of the material synthesis part repeatedly mentioned the same procedure for similar materials. As the authors themselves mentioned the procedure is well-known, thus it should be in brief and repeated procedures/sentences should be avoided.

2. The catalysts Pd/C-1 and Pd/C-2 were synthesized by taking 0.0172 M and 0.0155 M Pd-precursor solution with the constant amount (0.45 g) activated BAU-A, which shows 10% loading of Pd. Even with the diluted solution with the same Pd content, how is the same % of Pd up taken by the carbon? Elemental analysis by ICP/CHNO should be added.

3. Authors mentioned catalytic activity is highly dependent on the composition and nature of the catalysts. Thus, to support XRD, XPS data mentioning the composition of catalysts with respect to Pd0 and Pd2+ should be provided.

4. The Structure-activity relationship i.e. The role of Pd(OH)2/C) in the hydrogenation of quinoline and Pd/C in the hydrogenation of dienes in terms of selectivity should be elaborated with the proper mechanism.

5. I didn’t find any supporting information, as mentioned in the manuscript.

Author Response

Reviewer 1

COMMENT

In the present manuscript entitled “Screening of Palladium/Charcoal Catalysts for Hydrogenation of Diene Carboxylates with Isolated-Rings (Hetero)aliphatic Scaffold” authors have reported a series of Pd-carbon based catalysts, prepared by known procedures, for selective hydrogenation of diene carboxylates and, quinoline. The authors characterized the synthesized catalysts using various tools; and studied the hydrogenation reaction in-depth by varying reaction parameters. However, the manuscript still requires major modifications before acceptance for publication:

  1. The sub-sections of the material synthesis part repeatedly mentioned the same procedure for similar materials. As the authors themselves mentioned the procedure is well-known, thus it should be in brief and repeated procedures/sentences should be avoided.

REPLY

We agree with the Reviewer and we moved details of the composites preparation to the Supporting Information. Only the important details (the difference between the procedures) are discussed in the text of the revised manuscript, and general schemes are presented in the Materials and Methods section in the text of the revised manuscript.

COMMENT

  1. The catalysts Pd/C-1 and Pd/C-2 were synthesized by taking 0.0172 M and 0.0155 M Pd-precursor solution with the constant amount (0.45 g) activated BAU-A, which shows 10% loading of Pd. Even with the diluted solution with the same Pd content, how is the same % of Pd up taken by the carbon? Elemental analysis by ICP/CHNO should be added.

REPLY

The difference in the concentration of H2PdCl4 could have an influence only on the sorption kinetics at the beginning of the process, and could cause the difference in particles size. Total Pd content in the composites was the same (within the error of analysis), indicating that all Pd2+ from the solution was adsorbed. This finding can be explained by very high values of adsorption constant, as well as by partially irreversible nature of adsorption due to in situ reduction of Pd2+ to Pd0 and deposition of insoluble Pd nanoparticles (which occurs even without addition of reducer [Simonov, P.A.; Troitskii, S.Yu.; Likholobov, V.A. Preparation of the Pd/C Catalysts: A Molecular-Level Study of Active Site Formation. Kinet. Catal., 2000, 41, 255-269. https://doi.org/10.1007/BF02771428]).

The data of elemental analyses on Pd were added to the revised Supporting Information; determination of CHN content would be uninformative since the predominant content of carbon in the carrier. C content is (100 % - Pd content) and can be additionally shifted just by several % due to H and O, and, therefore, have insignificant influence on the catalytic activity.

COMMENT

  1. Authors mentioned catalytic activity is highly dependent on the composition and nature of the catalysts. Thus, to support XRD, XPS data mentioning the composition of catalysts with respect to Pd0 and Pd2+ should be provided.

REPLY

In fact, XPS data of the starting materials seem to have no direct relation to the catalytic activity of the Pd catalysts, because treatment of Pd catalyst with hydrogen leads to reduction of Pd2+: "There is only metallic Pd present at 1 bar H2 and 300 K irrespective of any precursor oxide" [Noack, K.; Zbinden, H.; Schlögl, R. Identification of the state of palladium in various hydrogenation catalysts by XPS. Catal. Lett., 1990, 4, 145-156. https://doi.org/10.1007/BF00765697]. In our study we performed hydrogenation at 100 bar of H2, and in these conditions all PdO had to be reduced to metallic Pd.

On the other hand, we used well-known catalysts, and their XPS (as well as XPS of many similar systems) were studied. Such composites always contain both metallic Pd and Pd2+. The state of Pd in the initial catalysts may have indirect relation to the catalytic properties, because reduction in situ results in formation of the metallic particles of different morphology. Expectedly, no systematic correlations between XPS-peak shapes and positions and the catalyst activity was observed [R. Burmeister, B. Despeyroux, K. Deller, K. Seibold, P. Albers. On the XPS-Surface Characterization of Activated Carbons resp. Pd/C Catalysts and a Correlation to the Catalytic Activity, in M. Guisnet et al. (Editors), Heterogeneous Catalysis and Fine Chemicals III, 1993, Elsevier Science Publishers B.V., page 361].

The nature of active site of Pd/C catalyst, which form in the course of hydrogenation process, depend also on the nature of substrate [Teschner, D.; Révay, Z.; Borsodi, J.; Hävecker, M.; Knop-Gericke, A.; Schlögl, R.; Milroy, D.; Jackson, S. D.; Torres, D.; Sautet, P. Understanding Palladium Hydrogenation Catalysts: When the Nature of the Reactive Molecule Controls the Nature of the Catalyst Active Phase. Angew. Chem. Int. Ed. 2008, 47, 9274-9278. https://doi.org/10.1002/anie.200802134]

Detailed study of such effects is interesting and complex task, however it is beyond the scope of this work.

We added respective comments on this issues with references to the revised manuscript.

COMMENT

  1. The Structure-activity relationship i.e. The role of Pd(OH)2/C) in the hydrogenation of quinoline and Pd/C in the hydrogenation of dienes in terms of selectivity should be elaborated with the proper mechanism.

REPLY

We suppose that under action of H2, PdO in Pearlman’s catalyst is reduced in situ to metallic Pd, which is a catalytically active species. Transformation of Pd2+ to the reduces species was shown in several studies [Noack, K.; Zbinden, H.; Schlögl, R. Identification of the state of palladium in various hydrogenation catalysts by XPS. Catal. Lett., 1990, 4, 145-156. https://doi.org/10.1007/BF00765697; Teschner, D.; Révay, Z.; Borsodi, J.; Hävecker, M.; Knop-Gericke, A.; Schlögl, R.; Milroy, D.; Jackson, S. D.; Torres, D.; Sautet, P. Understanding Palladium Hydrogenation Catalysts: When the Nature of the Reactive Molecule Controls the Nature of the Catalyst Active Phase. Angew. Chem. Int. Ed. 2008, 47, 9274-9278. https://doi.org/10.1002/anie.200802134]. Notably, apparently erroneous mechanisms of hydrogenation on Pearlman’s catalyst were proposed (for example, publication of 2010: https://doi.org/10.1002/9780470638859.conrr483), and such situation can be explained by the fact, that the structure of Pearlman’s catalyst was studied in details only in 2015 [Albers, P.W.; Möbus, K.; Wieland, S.D.; Parker, S.F.  The fine structure of Pearlman’s catalyst. Phys.Chem.Chem.Phys., 2015, 17, 5274-5278. https://doi.org/10.1039/c4cp05681g].

We added to the revised version of the manuscript a comment regarding in situ reduction of Pearlman’s catalyst and a comment regarding a difference between this catalyst and Pd/C species.

COMMENT

  1. I didn’t find any supporting information, as mentioned in the manuscript.

REPLY

The supporting information file was uploaded in the archive upon the initial submission of the original manuscript. We also upload the revised Supporting Information with submission of the revised manuscript. 

Reviewer 2 Report

Kolotilov and coworked reported the synthesis and characterization of various palladium-containing composites (four Pd/C and three Pearlman’s catalysts), testing their catalytic ability on the hydrogenation of three conjugated dienes, including tetrahydroquinoline. From their evidence, the preparation method of the composite has a more significant effect on its performance with respect to the Pd-nanoparticles size or their distribution on the charcoal surface. I think this manuscript is suitable for publication in Molecules, although a careful language review is suggested, but some minor points should be addressed:

-        a less conversational style should be utilized (e.g., “can’t” at page 3, line 114, should be changed into “cannot”).

-        Add “Figure 1” in the text. I would suggest after “pharmaceuticals” (page 2, line 59)

-        Page 3, line 119. What is Fig. 2a?

-        Table 2: N-methyl-1,2,3,4tetrahydroquinoline. N should be in italics.

-        Figure 10: N-Boc protected 5,6-dihydro-pyridine-1(2H)-carboxylates. N and 2H should be in italics.

-        Please, in the materials and methods, define the NMR multiplicity meaning, in particular, p (pentet?) and pd

Author Response

Reviewer 2

COMMENT

Comments and Suggestions for Authors

Kolotilov and coworked reported the synthesis and characterization of various palladium-containing composites (four Pd/C and three Pearlman’s catalysts), testing their catalytic ability on the hydrogenation of three conjugated dienes, including tetrahydroquinoline. From their evidence, the preparation method of the composite has a more significant effect on its performance with respect to the Pd-nanoparticles size or their distribution on the charcoal surface. I think this manuscript is suitable for publication in Molecules, although a careful language review is suggested, but some minor points should be addressed:

-        a less conversational style should be utilized (e.g., “can’t” at page 3, line 114, should be changed into “cannot”).

REPLY

The thank the Reviewer for this comment. A precise language review was performed; several rephrases and examples of conversational style were corrected throughout the manuscript.

COMMENT

-        Add “Figure 1” in the text. I would suggest after “pharmaceuticals” (page 2, line 59)

REPLY

The reference to the Figure was added, and Figure 1 was put below the section of the first mention.

COMMENT

-        Page 3, line 119. What is Fig. 2a?

REPLY

It was corrected to “Fig. 3” with a short specification.

COMMENT

-        Table 2: N-methyl-1,2,3,4tetrahydroquinoline. N should be in italics.

REPLY

It was corrected.

COMMENT

-        Figure 10: N-Boc protected 5,6-dihydro-pyridine-1(2H)-carboxylates. N and 2H should be in italics.

REPLY

It was corrected.

COMMENT

-        Please, in the materials and methods, define the NMR multiplicity meaning, in particular, p (pentet?) and pd

REPLY

“p” is an automatic Mestre software mark for quintet; it was corrected throughout the manuscript. A short part defining the NMR abbreviations was also added to the Materials and Methods section.

Round 2

Reviewer 1 Report

The manuscript can be considered for publication in its current form.